# A Review on SARS-CoV-2 Virology, Pathophysiology, Animal Models, and Anti-Viral Interventions

**DOI:** 10.3390/pathogens9060426

**Published:** 2020-05-29

**Authors:** Sabari Nath Neerukonda, Upendra Katneni

**Affiliations:** 1Department of Animal and Food Sciences, University of Delaware, Newark, DE 19716, USA; 2Current address: Department of Pediatrics, University of Maryland School of Medicine, Baltimore, MD 21201, USA

**Keywords:** SARS-CoV-2, COVID-19, animal models, antivirals

## Abstract

Severe acute respiratory syndrome coronavirus 2 (SARS-CoV-2), the causative agent of CoV disease 2019 (COVID-19) is a highly pathogenic and transmissible CoV that is presently plaguing the global human population and economy. No proven effective antiviral therapy or vaccine currently exists, and supportive care remains to be the cornerstone treatment. Through previous lessons learned from SARS-CoV-1 and MERS-CoV studies, scientific groups worldwide have rapidly expanded the knowledge pertaining to SARS-CoV-2 virology that includes in vitro and in vivo models for testing of antiviral therapies and randomized clinical trials. In the present narrative, we review SARS-CoV-2 virology, clinical features, pathophysiology, and animal models with a specific focus on the antiviral and adjunctive therapies currently being tested or that require testing in animal models and randomized clinical trials.

## 1. Introduction

By the end of December 2019, a novel coronavirus (2019-nCoV; later renamed SARS-CoV-2) was identified in Wuhan City of China’s Hubei province in 5 of 7 severe pneumonic patients admitted to the intensive care unit (ICU) of the Wuhan Jin Yin-Tan Hospital [1,2,3,4]. Metagenomics sequencing analysis of patients’ bronchoalveolar lavage fluid samples and de novo sequence assembly revealed 29.89kb genomes that shared 99.9%, 79.6%, and 96.2% sequence identity with each other, SARS-CoV-1 (BJ strain), and Bat CoV (RaTG13 strain), respectively [1]. Phylogenetic analysis of full-length genomes revealed the identity of SARS-CoV-2 as a newly emerged strain of betacoronavirus that formed a distinct lineage from other SARS-related CoVs (SARS-rCoVs) of bats, but closely related to RaTG13 [1]. Since its identification, the virus quickly spread within China and subsequently to 213 countries around the world within 3 months. By March 2020, WHO declared the disease caused by SARS-CoV-2 termed CoV disease 2019 (COVID-19) a pandemic. As of 28 May 2020, more than 5.9 million cases and 350,000 deaths had been reported globally, with these numbers still on the rise in many countries. In the United States alone, rapid community transmission resulted in more than 1.7 million cases and 100,000 deaths (Source: Johns Hopkins Coronavirus Resource Center; https://coronavirus.jhu.edu/).

## 2. Virology

### 2.1. Classification and Origin

SARS-CoV-2 is a member of the Coronavirinae subfamily in the family Coronaviridae and the order Nidovirales. Based on genetic and phylogenetic relationships, this subfamily is classified into four genera: α-, β-, γ-, and δ-CoV. Members of the α- and β-CoV genera infect mammals [5]. Members of the γ- and δ-CoV genera primarily infect birds and also some mammals [5]. CoVs are named after their crown-like morphology as observed under the electron microscope (EM) afforded by the surface glycoproteins on the virus. They are enveloped positive-sense single-stranded RNA viruses that can cause a wide range of respiratory, enteric, hepatic, renal, and neurologic diseases in birds and mammals [5]. Currently, circulating CoVs in humans include two α-CoVs (229E and NL63) and two β-CoVs (OC43 and HKU1) that cause common cold. The highly pathogenic human β-CoVs that emerged in the past two decades include SARS-CoV-1, the Middle Eastern respiratory syndrome coronavirus (MERS-CoV), and the more recent SARS-CoV-2 [5,6]. Bats are considered natural hosts for progenitors of highly pathogenic CoVs, whose spill over to humans is known to involve intermediate animal hosts [5]. While domestic civets and dromedary camels are known intermediate hosts of SARS-CoV-1 and MERS-CoV, respectively, the role of an intermediary host, if any, in SARS-CoV-2 spillover is presently unknown. A Pangolin-CoV that is closely related to RaTG13 (90.55% genome identity) and SARS-CoV-2 (91.02% genome identity) was identified in a dead Malayan pangolin which is now considered a possible intermediary host [7]. Human-to-human transmission via direct or indirect contact and inhalation of respiratory droplets are the main modes of global spread of highly pathogenic CoVs.

### 2.2. Virus Structure and Stability

SARS-CoV-2 virions are spherical, with a diameter of 60-100 nm conforming to the typical CoV diameter of 125 nm [6]. The virion surface displays characteristic club-shaped projections [8]. SARS-CoV-2 is highly stable at 4 °C for 2 weeks, whereas at room temperature, it can remain stable for a day without significant loss in infectivity [9,10]. It is sensitive to UV inactivation and heat treatment (56 °C for 30 min), with a 70 °C temperature reducing the virus inactivation time to 5 min. Its stability on printing paper or tissue paper (3 hr) is limited compared to wood and cloth (2 days) [9,10]. By contrast, it has higher stability (2–4 days) on smooth surfaces such as on banknotes, stainless steel, plastic, and both layers of a mask [9]. The stability and half-life of SARS-CoV-2 on stainless steel and plastic is comparable to SARS-CoV-1 [10]. Both these viruses have similar half-lives in aerosols as well, with a median estimate of 1.1–1.2 hrs [4]. SARS-CoV-2 remains sensitive to most disinfectants, including household bleach, povidone-iodine, chlorhexidine, chloroxylenol, benzalkonium chloride, diethyl ether, ethanol (70–75%), chlorine, peracetic acid, and chloroform [6,9].

### 2.3. SARS-CoV-2 Receptor and Entry

Like SARS-CoV-1, SARS-CoV-2 also utilizes human angiotenisin-converting enzyme 2 (ACE2) as a receptor for cellular entry [11]. ACE2 is a type I membrane glycoprotein expressed in lungs, heart, intestines, and kidneys [12]. Physiologically, ACE2 promotes cleavage and maturation of peptide hormones involved in the regulation of blood pressure, angiotenisin (Ang) I & II. ACE2 cleaves Ang I into Ang-(1-9), which then gets processed by other enzymes to generate Ang-(1-7). ACE2 can also directly process Ang II to generate Ang-(1-7) [12]. ACE2/Ang-(1-7) forms another arm of the renin–angiotensin system (RAS) and has cardiovascular protective effects, including anti-heart failure, anti-thrombosis, anti-myocardial hypertrophy, anti-fibrosis, anti-arrhythmia, and anti-atherogenesis [12]. Furthermore, ACE2 also protects from lung injury and is downregulated upon SARS-CoV-1 infection [13,14]. These findings ignited interest in ACE inhibitors and/or angiotensin receptor blockers for potential treatment of COVID-19. On the other hand, upregulation of ACE2 by these agents likely enhances viral entry and leads to worse disease outcomes [15]. Conflicting in vitro data exist regarding the benefit of these agents against SARS-CoV-2, but clinical practice guidelines recommend these agents for COVID-19 therapy [16].

Full-length ACE2 protein consists of an N-terminal peptidase domain (PD) and a C-terminal collectrin-like domain (CLD) followed by a single transmembrane helix and an ~40-residue intracellular segment [12]. The cryo-EM structure of SARS-CoV-2 spike (-S) trimer revealed metastable pre-fusion conformation which upon ACE2 binding undergoes substantial rearrangement to promote fusion between viral and cell membranes [8,17].

Unlike SARS-CoV-1-S, SARS-CoV-2-S harbors a multi-basic furin cleavage site (RRAR*SV) before the canonical S1/S2 site (AY*TM) which allows efficient proteolytic processing during viral egress and prior to the subsequent round of entry [18]. This processing is critical for the separation of the receptor-binding domain (RBD) and the fusion domain harbored in S1 and S2 subunits, respectively (Figure 1). The stochastic movement of the RBD in S1 transiently exposes its binding determinants in an “up” conformation that permits receptor binding. SARS-CoV-2-S binds ACE2 with a 10–20-fold higher affinity than SARS-CoV-1-S [8,17]. Receptor binding destabilizes the pre-fusion conformation, resulting in the shedding of the S1 subunit and transition of the S2 subunit into a highly stable post-fusion conformation [17].

Following binding and endocytosis, efficient proteolytic priming of S2 at a site proximal to the fusion peptide is essential for exposing the fusion peptide [11]. Like SARS-CoV-1-S, SARS-CoV-2-S either use endosomal cysteine proteases, cathepsin B and L (CatB/L), or the serine protease TMPRSS2 for S protein priming [11]. These proteases therefore are attractive drug targets. For example, inhibitors of CatB/L (E-64d) and TMPRSS2 (camostat mesylate, nafamostat mesylate) efficiently inhibit viral entry [11,19]. A Vero E6 cell line engineered to overexpress TMPRSS2 produces higher yields of SARS-CoV-2 compared to parent Vero E6 cell line [20]. However, TMPRSS2, but not CatB/L, is known to be essential for SARS-CoV-1 entry into primary target cells and for efficient viral replication and spread in the infected mice [21]. TMPRSS2 is co-expressed with the ACE2 receptor in multiple cells or tissues that are likely targets of SARS-CoV-2, including nasal goblet and ciliated cells, esophagus, ileum, colon, and cornea [22]. The relative importance of these proteases for SARS-CoV-2-S priming, viral pathogenesis, and target cell tropism need to be investigated in patient tissues and animal models. In the acidified endosomes, exposed fusion peptide inserts into the cell membrane, followed by the joining of two heptad repeats in S2′ into an anti-parallel six-helix bundle which facilitates mixing of viral and cellular membranes resulting in fusion [23]. Upon fusion, the genome is released into the cytoplasm.

### 2.4. General Features of Genome Replication and Particle Production

The SARS-CoV-2 genomic RNA contains a 5′ cap and a 3′ poly(A) tail that allow immediate translation to produce two coterminal replicase polyproteins, pp1a and pp1ab, by utilizing a ribosomal frameshifting mechanism. Replicase polyproteins are encoded within two-thirds of the genome (as ORF1a and ORF1b) (Figure 2) [1]. These polyproteins are subsequently cleaved by the action of two viral proteases (nsp3-PLpro and nsp5-Mpro) into the individual non-structural proteins (nsps), nsp1-11 and nsp1-16, respectively [23]. The remaining third codes for structural (spike (S), envelope (E), membrane (M), nucleoprotein (N)) and accessory proteins (ORF-3a, -3b, -6, -7a, -7b, -8, -9a, -9b, and 10; Figure 2) [1]. 

The nsps are assembled into the replicase–transcriptase complex which begins to generate anti-sense (-) genomic and sub-genomic RNAs that serve as templates for positive-sense genome (+) and sub-genomic mRNA synthesis, respectively. These sub-genomic mRNAs are 3′ co-terminal with the viral genome and form a nested set of RNAs due to discontinuous extension of the anti-sense RNA. The nsps also promote membrane rearrangement of rough endoplasmic reticulum (ER) membranes into double-membrane vesicles, in which the above replication and transcription processes occur [1].

Structural proteins translated from sub-genomic mRNAs are inserted into the ER and pass along the secretory pathway to the ER-Golgi intermediate compartment (ERGIC) [23]. Along this pathway, S protein is cleaved into two subunits, S1 and S2, by furin-like proteases [11]. The newly synthesized genome forms a complex with N proteinand bud into the ERGIC to generate virions. After the assembly, virions are transported to the cell surface in vesicles and are exocytosed [23]. Accessory proteins, also translated from sub-genomic mRNAs, carry out accessory functions that are likely important for pathogenesis and innate immune evasion [23]. Accessory protein functions of SARS-CoV-2 are yet to be characterized.

## 3. Clinical Features and Pathophysiology

Following transmission by droplets, contact, and other modes, the virus replicates in cells of respiratory (upper and lower) and gastrointestinal tracts. The clinical spectrum of COVID-19 can vary from asymptomatic and mild symptomatic to an adverse clinical condition characterized by severe respiratory failure requiring mechanical ventilation and hospitalization and critically ill condition [24]. Critically ill patients have systemic features characterized by sepsis, septic shock, and multiple organ dysfunction syndromes (MODS) [24]. Mild COVID-19 illness symptoms are typical of an upper respiratory tract viral infection, including mild fever, cough (dry), sore throat, nasal congestion, malaise, headache, muscle pain, but no dyspnea [24]. Infrequently, digestive symptoms such as nausea, vomiting, abdominal pain, diarrhea may be present.

Patients with moderate pneumonia display such symptoms as cough and shortness of breath (tachypnea). Patients clinically diagnosed with severe pneumonia display symptoms with or without fever, severe dyspnea, respiratory distress, tachypnea (> 30 breaths/min), and hypoxia (SpO2 < 90% on room air) [24]. Patients with sudden onset of respiratory failure or worsening of the already impaired lung function are diagnosed with acute respiratory distress syndrome (ARDS) and require mechanical ventilation [24]. Different ARDS forms include mild, moderate, and severe. Chest computerized tomography (CT) scans reveal bilateral ground glass opacities and subsegmental areas of consolidation [24].

Clinical features of COVID-19 patients with sepsis are particularly critical with a wide range of signs and symptoms involving multiple organs [24]. These signs and symptoms include respiratory manifestations, such as severe dyspnea and hypoxemia, kidney impairment with reduced urine output, tachycardia, altered mental status, and coagulation dysfunction. Functional alteration of multiple tissues as reflected in laboratory findings include lymphopenia, hyperbilirubinemia, increased D-dimers, acidosis, high lactate, coagulopathy, and thrombocytopenia.

Physiologically, patients with severe disease have significantly lower counts of T cells, CD8+ T cells and NK cells compared to those with mild disease [25]. Overall, COVID-19 patients have fewer CD8+ T cells and NK cells compared to healthy controls. In addition, NK- and CD8^+^ T-cell populations in COVID-19 patients are functionally exhausted (NKG2A^+^), whereas convalescing patients recover functionality [25]. Out of 41 SARS-CoV-2-positive pneumonic patients admitted to Jin Yin-tan Hospital, 16 patients progressed to ARDS and were therefore placed under ICU care [26]. Both ICU and non-ICU patients had higher initial plasma concentrations of IL-1B, IL-1RA, IL-7, IL-8, IL-9, IL-10, basic FGF, G-CSF, GM-CSF, IFN-γ, IP-10, MCP-1, MIP-1A, MIP-1B, PDGF, TNF-α, and VEGF compared to healthy adult controls, whereas IL-5, IL-12p70, IL-15, eotaxin, and Regulated on Activation Normal T cell Expressed and Secreted (RANTES) were similar [26]. A comparison between ICU and non-ICU patients showed higher plasma concentrations of IL-2, IL-7, IL-10, G-CSF, IP-10, MCP1, MIP1A, and TNFα in ICU patients than in non-ICU patients substantiating the association of “cytokine storm” in severe disease [26].

A separate study also reported decreased counts of T cells and monocytes in COVID-19 patients compared to healthy controls [27]. While CD4+ T cells displayed a remarkable decrease in both ICU and non-ICU patients, decrease in CD8^+^ T cells was noted only in ICU patients. COVID-19 patient CD4^+^ and CD8^+^ T cells displayed activated phenotype with higher expression of CD69, CD38, and CD44 [27]. CD4^+^ T cells of ICU patients with more severe pneumonia showed higher percentage of GM-CSF^+^ and IL-6^+^ cells compared to non-ICU patients or healthy adults. A higher percentage of CD14^+^CD16^+^ inflammatory monocytes in peripheral blood of COVID-19 patients with a much higher percentage in ICU patients were noted [27]. Notably, these inflammatory monocytes were GM-CSF^+^ and IL-6^+^. Higher percentages of activated CD4^+^ T cells and inflammatory monocytes in severe patients accompanied with GM-CSF and IL-6 production likely contribute to cytokine storm and pulmonary immune pathology in these patients. Interestingly, no differences in granulocyte, B cell and NK cell counts were noted between COVID-19 and healthy patients in this study [27].

A separate study on 452 patients admitted to Tongji Hospital, China, also reported higher leukocyte counts, neutrophil counts, lower lymphocyte counts, higher neutrophil-to-lymphocyte ratio, as well as a lower percentage of monocytes, eosinophils, and basophils in severe patients compared to non-severe ones [28]. Severe patients also had higher levels of IL-2R, IL-6, IL-8, IL-10, and TNF-α, but no differences in IgA, IgG, and complement proteins (C3 or C4). Percentage reduction in CD28^+^ cytotoxic suppressor T cells (CD3^+^CD8^+^CD28^+^) and regulatory T cells (CD3^+^CD4^+^CD25^+^CD127^low+^), was observed in severe cases [28]. Overall, immune dysregulation and exuberant inflammatory response in severe patients is due to higher neutrophils, activated monocytes and/or T cells, and reduced suppressor or regulatory T cells [28].

## 4. Animals Models of COVID-19

Robust animal models that faithfully recapitulate SARS-CoV-2 pathogenesis as in humans do not exist as of now. Widely used laboratory mice are less susceptible to the SARS-CoV-2 infection, likely due to mouse ACE2 receptor incompatibility with S glycoprotein. However, research tools such as a mouse-adapted (MA) SARS-CoV-2 strain and human ACE2 (hACE2) transgenic mice may overcome the incompatibility issue. Nevertheless, C57BL/6 and BALB/c mice infected with SARS-CoV-2 displayed transient infection and viral replication that is limited by an induction of innate immune response [29]. However, SARS-CoV-2 underwent efficient replication in the lower respiratory tract of hACE2-transgenic mice, but only caused mild or moderate interstitial pneumonia with no major clinical symptoms [30]. Another group recently demonstrated altered viral tropism and pathogenesis in SARS-CoV-2-infected hACE2 transgenic mice. Although hACE2 transgenic mice displayed active viral replication in lungs, 40% mice mortality was observed by 5 days post infection (dpi) due to brain tropism [31]. To overcome altered tropism and more accurately model the respiratory disease observed in humans, the authors adapted S RBD interaction with mouse ACE2 through directed evolution of RBD interaction residues (Q498T/P999Y) [31]. This MA SARS-CoV-2 displayed replication in upper and lower respiratory tracts and caused no mortality. Age-dependent pathogenesis as in humans was seen with infected 12-week-old and 1-year-old female BALB/c mice displaying mild or moderate and severe disease, respectively [31]. This model was successfully used in MA SARS-CoV-2 prophylaxis and therapy employing Venezuelan equine encephalitis virus (VEEV)-replicon and pegylated IFN-λ [31].

By contrast, SARS-CoV-2-infected Syrian hamsters displayed early active virus replication in lungs and marked lung pathology characterized by necrotizing bronchiolitis, massive leukocyte infiltration, and edema, recapitulating pathological changes of SARS-CoV-1-infected Syrian hamsters and severe pneumonic patients [29]. In addition, a double-edged role of STAT2 was demonstrated in the SARS-CoV-2 hamster model [29]. STAT2 knockout (KO) hamsters displayed higher lung titers, viremia, and systemic spread compared to wild-type ones indicating the role of STAT2 in limiting systemic spread [29]. On the other hand, limited leukocyte infiltration, attenuated lung pathology, and no pneumonia was observed in STAT2 KO hamsters.

Ferrets have been successfully used to model respiratory disease and transmission of several respiratory viruses, including influenza, respiratory syncytial virus, parainfluenza viruses, and SARS-CoV-1 [32]. Similarities in tissue architecture and anatomy of ferret respiratory system to human with respect to receptor distribution, submucosal gland density in the bronchial wall, and the number of terminal bronchioles made them attractive models of study. Susceptibility of various laboratory, companion, and domestic animals to SARS-CoV-2 has been tested to model pathogenic mechanisms and the efficacy of control interventions. Several animal models that were tested for SARS-CoV-2 susceptibility include 3-4-month-old female ferrets, 12-20-month-old mixed sex ferrets, sub-adult or juvenile cats, 3-month-old beagles, 40-day-old mixed sex specific-pathogen-free (SPF) Landrace and Large White pigs, 4-week-old specific-pathogen-free (SPF) White Leghorn chickens, and 4-week-old SPF ducks [33].

SARS-CoV-2 also underwent efficient viral replication in organs of the upper respiratory tract of 3-4-month-old infected or contact ferrets, including nasal turbinate, soft palate, and tonsils, while neither virus nor viral replication was detected in the trachea, lungs, heart, kidneys, spleen, pancreas, small intestine, and brain [33]. In contrast, in 12-20-month-old mixed sex ferrets that are infected or contact-exposed to SARS-CoV-2, efficient replication was observed in organs of the upper or lower respiratory tract along with viral RNA detection in trachea, kidneys, and intestines [33]. Furthermore, efficient viral shedding was noticed in biological fluids (saliva, urine, and nasal washes) and feces of infected/contact ferrets as observed in humans [33]. Apart from age-related difference in pathogenesis, one limitation of SARS-CoV-2-infected ferrets is lower lung virus titers with no severe disease or mortality compared to other animal models, such as SARS-CoV-1-infected or MERS-CoV-infected hACE2 or hDPP4 transgenic mice [33].

Age-related difference in viral replication was also noted among cats with juvenile cats being more permissive compared to sub-adult cats [33]. Efficient viral replication was noted in all the above organs tested in juvenile cats, whereas in sub-adult cats, viral replication was limited to organs of the upper respiratory tract [33]. Until now, companion cats and large cats in zoos were reported positive due to human-to-cat transmission with the former displaying respiratory signs [34]. Among other SARS-CoV-2 infected animals, dogs displayed minor susceptibility, whereas livestock, including pigs, chickens, and ducks, displayed no susceptibility [33]. In line with this, two companion dogs of COVID-19 patients, one old age Pomeranian with several pre-existing conditions and an adult German Shepherd tested positive for viral RNA in nasal and oral swabs with the latter demonstrating higher viral RNA loads [35]. Dog viral genomic RNA matched with respective patient transmitters. The potential for reservoir and onward transmission by cats or dogs is presently unknown and requires investigation [35].

Recently, non-human primate models, including young and aged cynomolgus macaques and young rhesus macaques, were investigated as potential animal models of COVID-19 [36,37,38]. All SARS-CoV-2-infected macaques displayed productive infection with clinical signs ranging from asymptomatic in cynomolgus macaques to moderate disease in rhesus macaques [36,37,38]. Only rhesus macaques displayed transient respiratory disease and weight loss [37]. Peak early viral replication in the respiratory tissues was followed by a rapid decline, although aged cynomolgus and young rhesus macaques continued to shed the virus for prolonged periods. Viral antigen was detected in nasal epithelium, ciliated epithelial cells, type I and type II pneumocytes, bronchial and bronchiolar epithelial cells and some macrophages [36,37,38]. Injury to these cell types correlated with diffuse alveolar damage, alveolar and bronchiolar epithelial necrosis, alveolar edema, hyaline membrane formation, infiltration of neutrophils, macrophages and lymphocytes, and interstitial pneumonia [36,37,38]. While non-human primate models fail to recapitulate severe disease and critical condition observed in COVID-19 patients, they may be useful for the evaluation and licensure of preventive and therapeutic strategies [36,37,38]. A DNA immunogen expressing full-length S immunogen provided optimal protection in SARS-CoV-2 rhesus macaque challenge model compared to soluble constructs and smaller fragments of S immunogens [39]. Protection efficacy correlated with reduced viral replication in upper and lower respiratory tracts and neutralization antibody titers. Non-human primate models, however, come at an increased expense [39].

## 5. Anti-Viral Interventions

Current antiviral interventions for COVID-19 are mostly learned from SARS-CoV-1 and MERS-CoV studies and include antiviral therapeutics and various vaccine platforms. Lessons learned from SARS-CoV-1 and MERS-CoV vaccine studies indicated the S protein on the surface of the virus to be an ideal target for vaccine preparation [40,41,42]. Antibodies targeting the S can interfere with S-ACE2 interactions and therefore neutralize the virus. Several vaccine platforms targeting the S protein are in pipeline, including RNA-, DNA- recombinant protein- and viral vector-based vaccines that are beyond the scope of this review and reviewed elsewhere [43]. For the purpose of this review, we tend to focus on antiviral therapies and their in vitro and in vivo testing status.

In order to be effective against emerging CoVs, antiviral therapeutics must target viral or host factors that are 1) highly conserved among CoVs of diverse hosts, including reservoirs (e.g., human, bat, pig, etc.), and 2) essential to viral replication or pathogenesis. Targeted inhibition of highly conserved viral or host factors involved in coronavirus lifecycle will likely result in broad spectrum antivirals that ameliorate disease signs and pathology by reducing viral loads and modulation of host immune response (Figure 3). In this regard, CoV nsps represent attractive targets, as they are highly conserved compared to structural or virus-specific accessory proteins and carry out essential replication functions. Additionally, host proteins that are usurped by viral proteins via direct or indirect interactions to mediate critical functions of replication and pathogenesis also represent potential druggable targets as described recently in SARS-CoV-2 interactome study [44]. As described below, several broad-spectrum antivirals previously identified against SARS-, MERS-, and other bat CoVs have a promising antiviral activity against emerging SARS-CoV-2.

### 5.1. Anti-Viral Therapeutics

#### 5.1.1. β-D-N4-hydroxycytidine (NHC)

NHC and its orally available prodrug (EIDD-2801) are ribonucleoside analogs with broad spectrum antiviral activity against multiple RNA viruses, including the current SARS-CoV-2, SARS-CoV-1, MERS-CoV, Venezuelan equine encephalitis virus (VEEV), influenza A/B, Ebola (EboV), and chikungunya (CHIKV) viruses [33]. In vitro, low micromolar concentrations of NHC induced a potent antiviral effect towards SARS-CoV-2 [EC_50_: 0.08-0.3 μM], MERS [EC_50_: 0.15 μM], and SARS-CoV-1 [EC_50_: 0.14 μM] propagated in Calu-3 cell lines and primary human airway epithelial cultures [33]. In vivo, prophylactic and therapeutic treatment with EIDD-2801 significantly reduced lung viral loads and improved pulmonary function in SARS- and MERS-CoV mouse models. However, therapeutic improvement diminished with a delay in treatment initiation time [33]. Thus, an early intervention with EIDD-2801 may significantly improve the clinical outcome of COVID-19 patients which deserves future investigation.

Mechanism of action (MOA) of NHC/EIDD-2801 is by targeting viral RNA-dependent RNA polymerase (vRdRp) to induce error catastrophe beyond the error threshold allowed to sustain RNA virus quasispecies [33]. NHC incorporation as a C or a U in an anti-sense genome or positive-strand genomic/subgenomic mRNA results in increased nucleotide transitions (A to G, G to A, C to U, U to C) and therefore non-synonymous substitutions. A positive correlation between the increased transition rate, the frequency of non-synonymous mutations, and therapeutic efficacy was observed in mice. While MOA of remdesivir (RDV) is via chain termination, it can generate resistance (see below). Resistant mutants of RDV are still inhibited by NHC [33].

#### 5.1.2. Remdesivir

Remdesivir (RDV) is a prodrug of the C-adenosine nucleoside analog, 1-cyano 4-aza-7,9-dideazaadenosine C-nucleoside with broad spectrum antiviral activity against filoviruses, paramyxoviruses, pneumoviruses, and CoVs, including SARS-CoV, MERS-CoV, mouse hepatitis virus (MHV), OC43, 229E, porcine, and bat CoV strains [45,46,47]. RDV, similar to other nucleoside analogs (favipiravir, ribavirin, and EIDD-2801), is converted from its pro-form to the active triphosphate form which inhibits vRdRp by non-obligate RNA chain termination [48]. RDV is incorporated into nascent viral RNA chains by the vRdRp to result in chain termination [49]. It displayed potent in vitro antiviral activity towards SARS-CoV-2 with an EC_50_ of 0.77 μM in Vero E6 cells [45]. In vitro, remdesivir also displayed potent antiviral activity towards SARS-CoV-1 [EC_50_: 0.069 μM] and MERS-CoV [EC_50_: 0.074 μM] at low micromolar concentrations [47].

In non-human primate (NHP) model of EboV disease, intravenous (i.v.) administration of 10 mg/kg of RDV resulted in persistent active concentration of 10 μM in blood which conferred 100% protection against EboV infection. Effective serum concentration of RDV is therefore likely achieved in NHP and human models. In a mouse model of SARS-CoV-1 disease, prophylactic and early therapeutic treatment with 1 dpi caused a reduction in lung viral replication with no defects in pulmonary function [47]. Therapeutic treatment at 2 dpi also reduced lung viral replication, although it did not improve the pulmonary function [47]. Similarly, in a humanized DPP4 transgenic mouse with improved pharmacokinetics for nucleotide prodrugs (Ces1c^−/−^ hDPP4), prophylactic and therapeutic treatment with RDV reduced lung viral replication and weight loss and improved pulmonary function [50]. Furthermore, in this mouse model, RDV provided superior antiviral activity against MERS-CoV compared with LPV/RTV-IFN-β combination (see below) [50]. In both SARS- and MERS-CoV disease models, although therapeutic treatment initiated after peak lung viral replication and lung damage led to reduced viral loads, it failed to reduce lung pathology or clinical disease. In a macaque model of MERS-CoV disease, prophylactic treatment successfully reduced viral loads and lung pathology, whereas therapeutic treatment during peak viral loads reduced viral loads and lung pathology in 2 out 6 animals [51]. Hence, clinical benefit of RDV is likely limited to prophylaxis and early therapy before viral loads reach high titers [51]. Resistant variants to RDV were identified through serial passage of MHV in the presence of increasing concentration of RDV which bore two mutations in vRdRp (F480L+V557L) [49]. Both mutations also conferred resistance in SARS-CoV-1 indicative of conserved mechanism of action (MOA) towards CoVs. However, SARS-CoV-1 resistance to RDV came with a fitness cost in vivo resulting in attenuated pathogenesis in a mouse model [49].

Case reports describing successful use of RDV for COVID-19 treatment exist and RDV is the most promising drug so far for COVID treatment [52,53]. Many randomized clinical trials (RCTs) are presently evaluating the safety and antiviral activity of RDV in patients with mild to moderate or severe COVID-19 (NCT04257656, NCT04252664, NCT04292730, NCT04292899, NCT04280705) [16]. RDV is not currently FDA-approved and must be procured for compassionate use only (for children <18 y.o. and pregnant women), expanded access, or clinical trial enrollment [16]. However, FDA has recently issued an emergency use authorization (EUA) for emergency use of RDV for the treatment of hospitalized COVID-19 patients upon review of data from the Gilead-sponsored open-label trial (NCT04292899) and an RCT carried out by NIAID (NCT04280705) [54].

#### 5.1.3. Ivermectin

Ivermectin is an FDA-approved anti-parasitic drug that has been known to exhibit broad spectrum antiviral activity against HIV, VEEV [55], dengue virus (DENV) [56], Yellow fever virus (YFV), Japanese encephalitis virus (JEV), West Nile virus (WNV), Tick borne encephalitis virus (TBEV) [57], and influenza [15]. Ivermectin targets both viral and host factors. For instance, ivermectin is known to inhibit classical nuclear transport of viral proteins (e.g., DENV NS5) relying on importin α/β heterodimer [58]. As viruses usurp host nuclear transport machinery for transport of viral and cellular factors, targeting host machinery is an attractive antiviral approach as the genetic barrier for resistance is much higher for the host compared to the virus [59]. On the other hand, ivermectin also acted as an uncompetitive helicase inhibitor of flavivirus NS3 helicase and exhibited antiviral effect against all flaviviruses tested [57]. Recently, ivermectin has been found to exhibit antiviral activity against SARS-CoV-2 [EC_50_: 2 μM] at low micromolar concentration in vitro with no cytotoxic effects [60]. Although it specifically inhibited viral genome replication, potential MOA of this finding remains to be investigated.

#### 5.1.4. N3

By structure-based ab initio drug design combined with virtual screening and high-throughput screening of drugs including the FDA-approved drugs, clinical trial drugs, and others, two compounds, namely, N3 and ebselen, were identified to have antiviral activity towards SARS-CoV-2 [61]. Ebselen and N3 displayed in vitro antiviral activity by inhibiting SARS-CoV-2 main protease (Mpro) with EC_50_ values of 4.67 μM and 16.77 μM, respectively [61]. Non-covalent binding and irreversible covalent modification of Mpro were proposed MOA of ebselen and N3, respectively. Therapeutic efficacy of N3 and ebselen against SARS-CoV-2 needs to be investigated in animal models either alone or in combination therapy.

#### 5.1.5. Lopinavir/Ritonavir

Lopinavir (LPV) and ritonavir (RTV) are FDA-approved HIV-1 protease inhibitors that also appear to inhibit 3CLpro activity in vitro [62,63]. Lopinavir has poor bioavailability, but it is more potent in inhibiting HIV-1 than ritonavir [64]. As part of HIV medication, lopinavir is used in combination with ritonavir which inhibits the lopinavir metabolizing enzyme, cytochrome P450 3A4, and improves its bioavailability [65]. LPV, but not RTV, was shown to have antiviral effect towards SARS-CoV, MERS-CoV, and 229E-GFP with EC_50_ concentrations of 17.1 μM, 8 μM, and 6.6 μM respectively. However, LPV displayed no effect towards MERS-CoV in Vero cells but in Huh7 cells [63]. Similarly, LPV, but not RTV, also demonstrated antiviral effect towards SARS-CoV-2 [logEC_50_: 26.1 μM] in vitro. Furthermore, LPV/RTV in combination with IFN-β displayed no synergistic antiviral activity against MERS-CoV compared to IFN-β alone [EC_50_ = 160 vs 175 IU/mL, respectively] in Calu-3 cells. The same study also demonstrated near similar EC_50_ for LPV/RTV [EC_50_: 8.5 μM] and LPV [11.6 μM], suggesting similar activity as observed for SARS-CoV-1.

Despite in vitro antiviral activity against MERS-CoV, prophylactic or therapeutic administration of LPV/RTV + IFN-β in a mouse model failed to reduce the virus titer, prevent body weight loss / pulmonary function / inflammatory lung disease. In contrast, in a common marmoset model of severe MERS-CoV disease, LPV/RTV treatment resulted in reduced lung / extrapulmonary tissue viral loads and improved the clinical, radiological, and pathological features compared to untreated and mycophenolate mofetil-treated marmosets. The MIRACLE trial (MERS-CoV Infection tReated with A Combination of Lopinavir/ritonavir and intErferon-β1b) is presently investigating the efficacy of combination therapy of LPV/RTV + IFN-β compared to placebo in MERS-confirmed hospitalized patients provided with standard supportive care [66].

In a non-RCT comprising SARS-CoV-1 patients, efficacy of LPV/RTV in combination with ribavirin was compared to ribavirin and steroids. LPV/RTV with or without ribavirin treatment induced a milder disease course in terms of diarrhea, fever, and chest radiographs and a reduction in the number of patients undergoing ARDS and death [62]. Notably, the clinical benefit of LPV/RTV was only demonstrated in patients that received initial treatment with LPV/RTV that is the treatment initiated at the time of SARS-CoV-1 diagnosis. Currently, the efficacy of LPV/RTV with or without ribavirin is also being evaluated in SARS-CoV-2 patients under randomized control trials [16]. Although the minimum attainable serum concentration of LPV/RTV is less than the minimum in vitro EC_50_ of SARS-CoV-2, LPV/RTV with or without ribavirin at 400 mg / 100 mg twice daily is part of the recommended treatment for COVID-19 patients in China [16]. An open-label RCT reported no significant benefit of lopinavir-ritonavir (400/100 mg twice daily) in SARS-CoV-2 patients than the standard care in terms of time to clinical improvement, mortality at 28 days, and throat swab viral loads at various time points [67]. Furthermore, gastrointestinal adverse events, such as nausea, vomiting, and diarrhea, appeared more common in the LPV/RTV group than in the standard care group [67]. Although delayed treatment initiation may partially explain the lack of clinical benefit of LPV/RTV, a subgroup analysis did not find reduced time to clinical improvement for patients who received LPV/RTV within 12 days.

In Singapore, 5 of 18 earlier SARS-CoV-2 patients on supplemental oxygen were treated with LPV/RTV. The fever resolved and supplemental oxygen requirement was reduced within 3 days in 3 of 5 patients, whereas 2 patients developed progressive respiratory failure. Four of 5 patients developed nausea, vomiting, and/or diarrhea, and 3 had abnormal liver function [68]. In Korea, one patient treated with LPV/RTV starting day 10 of illness had reduced viral loads which eventually remained undetectable [69]. Two case studies comprising Chinese patients also reported improved clinical outcomes upon combination therapy with LPV/RTV + arbidol + ShufengJiedu Capsule (SFJDC, a traditional Chinese medicine) and LPV/RTV (800/200 mg daily) + methylprednisolone (40 mg daily) + IFNα-2b (10 million IU daily) + ambroxol HCl [70,71]. It is difficult to interpret the outcomes of these studies due to additional drug therapies, varied time points of initial therapy, varied degree of severity of illnesses, and lack of control or placebo treatment. Given the significant drug–drug interactions and potential adverse effects (hepatotoxicity), cautious review of co-administered drugs and monitoring is required prior to use with LPV/RTV [64]. With the above data, LPV/RTV has limited advantage over the standard care for SARS-CoV-2 patients [59]. More importantly, whether an early therapy initiation leads to improved clinical outcomes compared to rescue and/or salvage therapy warrants investigation [16].

Another second-generation HIV-1 protease inhibitor darunavir demonstrated in vitro antiviral activity against SARS-CoV-2 [EC_50_: 300 μM] [72]. Darunavir treatment inhibited SARS-CoV-2 replication by 280-fold compared to non-treated control [72]. An RCT of darunavir/cobicistat is currently underway in china [16].

#### 5.1.6. Chloroquine and Hydroxychloroquine

Chloroquine is an anti-malarial drug with anti-inflammatory and immunomodulatory properties that is being investigated as a potential therapeutic for SARS-CoV-2-infected patients [16]. Chloroquine prevents endosomal acidification required for S protein conformational changes and thereby blocks fusion of viral and cellular membranes [16]. In addition, chloroquine also interferes with glycosylation of viral entry receptors. Chloroquine demonstrated potent in vitro activity against SARS-CoV-2 [EC_50_: 1.13 μM] at low micromolar concentration in Vero E6 cells [45,62,73]. This concentration is within the EC_50_ range reported for SARS-CoV-1 [EC_50_: 1–8.8 μm] and MERS-CoV [EC_50_: 3.0 μM] in Vero E6 and Huh7 cell lines, respectively [63]. These findings have prompted the use of chloroquine in numerous clinical trials with COVID-19 patients. Gao et al. reported that “thus far, results from more than 100 patients have demonstrated that chloroquine phosphate is superior to the control treatment in inhibiting the exacerbation of pneumonia, improving lung imaging findings, promoting a virus-negative conversion, and shortening the disease course according to the news briefing [74]. Severe adverse reactions to chloroquine phosphate were not noted in the aforementioned patients.”

Although these preliminary results were promising, drug supply issues and cardiotoxicity concerns limited the use of chloroquine in the US [64]. An alternative to chloroquine known as hydroxychloroquine is being pursued for the effective treatment of COVID-19 patients. Hydroxychloroquine [EC_50_: 0.72 μM] exhibited a 7.6-fold greater potency than chloroquine [EC_50_: 5.47 μM] in SARS-CoV-2-infected Vero cells [75]. In contrast, chloroquine [EC_50_: 6.5 ± 3.2 μM] exhibited approximately a 5-fold greater potency compared to hydroxychloroquine [EC_50_: 34 ± 5 μM] in SARS-CoV-1-infected Vero cells [73].

A non-randomized clinical trial comprising 36 patients (20 hydroxychloroquine and 16 control) aged above 12 and PCR+ for SARS-CoV-2 was conducted to evaluate the therapeutic efficacy of hydroxychloroquine. Viral clearance in nasopharyngeal swabs was considered the primary endpoint. By day 6, 70% (14/20) and 12.5% (2/16) of hydroxychloroquine and control patients, respectively, had viral clearance [76]. In the same study, addition of azithromycin to hydroxychloroquine in 6 patients resulted in superior viral clearance (6/6, 100%) compared with hydroxychloroquine alone (8/14, 57%) [76]. In another prospective trial, 30 randomized patients were split in a 1:1 fashion and treated with hydroxychloroquine daily for 5 days + the standard care (supportive care, IFN, and other antivirals) or the standard care alone [77]. No difference in virological clearance was noted, with 86.7% and 93.3% clearance observed for the hydroxychloroquine + the standard care group and the standard care group, respectively. Limited sample size and cardiotoxicity concerns call for additional studies before adopting this regimen for therapy. An observational study of effectiveness of chloroquine and hydroxychloroquine reported a lack of higher or lower risk of intubation or death among patients [78]. There are several RCTs presently examining the efficacy of chloroquine and hydroxychloroquine in COVID-19 patients [16,64].

#### 5.1.7. Type I IFN

CoV infections are typically associated with attenuated innate interferon induction and delayed pro-inflammatory response. In HAE cultures, SARS-CoV-1, MERS-CoV, and HCoV-229E infections resulted in no earlier transcriptional induction of interferons and pro-inflammatory cytokines at 3, 6, and 12 hours post infection (hpi) [79]. In Calu-3 cells, unlike MERS-CoV, SARS-CoV-1 and -229E, the infection caused slightly higher induction of IFN-β at 30 hpi [79]. MERS-CoV, however, is highly sensitive to the antiviral effects of IFN, unlike SARS-CoV-1 in vitro. Antiviral activity of IFN-β (EC_50_ = 1.37 units/mL) is 16-, 41-, 83- and 117-fold higher than of IFN- α2b, IFN-γ, IFN-universal type 1, and IFN-α2a, respectively [79]. Like MERS-CoV, SARS-CoV-2 is highly sensitive to the antiviral effects of recombinant type I IFN in Vero cells with a 30–40-fold reduction in viral titers [80]. Although SARS-CoV-2 blocked early induction of type I IFN, a significant induction of IFN and downstream STAT1 phosphorylation was noticed by 48 hpi in Calu-3 cells [80]. The reduced capacity of SARS-CoV-2 to interfere with induction and action of type I IFN are attributed to the loss of ORF3b and truncation/changes in ORF6, respectively [80]. For SARS-CoV-1, ORF3b and ORF6 disrupt IRF3 phosphorylation and karyopherin function, respectively, resulting in the attenuated induction of type I IFN and blocked nuclear transport of STAT1 [80].

In common marmosets infected with MERS-CoV, clinical improvements with IFN-β were similar to the LPV/RTV ones described above [81]. Similar to in vitro findings, loss of type I IFN signaling had no significant impact on SARS-CoV-1 disease in vivo, indicative of its insensitivity to the antiviral effects of type I IFN [82]. An interventional study to evaluate the efficacy and safety of LPV/RTV in adult hospitalized patients with SARS-CoV-2 infection is being pursued in China [64]. Due to the lack of clinical trial data and conflicting in vitro and in vivo data, interferons are not currently recommended for SARS-CoV-2 treatment [64].

#### 5.1.8. Arbidol

Arbidol (umifenovir) is a more promising repurposed drug which targets the S protein/ACE2 interaction, thereby inhibiting viral fusion and entry [16]. Although not FDA-approved, it is currently approved in Russia and China for the treatment and prophylaxis of influenza and has in vitro antiviral activity against SARS-CoV-2 [EC_50_: 10–30 μM] [83]. A non-randomized study of 67 COVID-19 patients indicated that treatment with arbidol for a median duration of 9 days was associated with lower mortality (0% (0/36) vs 16% (5/31)) and higher discharge rate compared with patients who did not receive it [84]. There are ongoing RCTs that are currently evaluating arbidol for COVID-19 treatment in China.

#### 5.1.9. Ribavirin

Ribavirin is a guanosine analog with anti-viral effects against several viral infections including respiratory syncitial virus (RSV), hepatitis C virus (HCV), Herpes Simplex Virus (HSV)-1. Ribavirin displayed no antiviral effect towards MERS-CoV and SARS-CoV-2 in vitro at concentrations below 100 µM in Vero E6 cells [45]. Another study reported conflicting findings where ribavirin displayed antiviral effects [logEC_50_: 9.99 μM] towards MERS-CoV replication in Vero E6 cells [85]. Vero cells are comparably inefficient at converting ribavirin into its mono- and tri-phosphate forms, therefore not a relevant cell line to study the antiviral effects of nucleoside analogs, including ribavirin [86]. A separate study identified better responsiveness of MERS-CoV-infected LLC-MK2 cells to the antiviral effects of ribavirin [EC_50_: 16.33 μg/mL] compared to Vero cells [EC_50_: 41.45 μg/mL] [87]. Similarly, sensitivity of SARS-CoV-1 to ribavirin also appears to be cell line-dependent, with higher concentration [EC_50_: 50 μg/mL] reported to be effective in Vero cells [87]. However, these concentrations exceeded the attainable serum concentrations in humans and therefore, ribavirin use may not be effective towards CoV infections in human [64].

A combination of IFN-α and ribavirin was previously reported to have synergistic and additive effect towards SARS- and MERS-CoV infections, respectively. This combination also reduced virus replication, downregulated host inflammatory response, and improved clinical outcome in MERS-CoV-infected macaques [88]. However, in a small-scale clinical trial comprising 5 critically ill MERS patients with multiple comorbidities, delayed initiation of combination therapy on or after the 19^th^ day of admission resulted in fatal outcome in all patients [89]. In addition, ribavirin usage is limited in patients with respiratory disorders, as it can lead to a marked fall in hemoglobin levels and anemia [64]. In line with this, a meta-analysis on SARS-CoV-1 and MERS-CoV patient case studies has found limited (if any) efficacy of ribavirin in outcomes of patients with highly pathogenic coronavirus respiratory syndromes. Due to the lack of in vitro efficacy, toxicity profile, and poor clinical outcomes, ribavirin was not considered a promising candidate for further evaluation by the WHO research and development plan for SARS-CoV-2 [64].

#### 5.1.10. Nizatoxanide

Nitazoxanide [2-acetyloxy-N-(5-nitro-2-thiazolyl) benzamide] is an FDA-approved broad-spectrum anti-parasitic compound effective against intestinal protozoal and helminthic infections, specifically, *Giardia lamblia* and *Cryptosporidium parvum* [90]. Recently, this compound has been shown to have a broad spectrum antiviral activity against multiple viruses, including influenza virus [91], rotavirus [92], astrovirus [93], norovirus [94], Japanese encephalitis virus (JEV) [95], rubella virus [96], Zika virus (ZIKV) [97], hepatitis C virus (HCV) [98], and hepatitis B virus (HBV) [99]. Nitazoxanide displayed potent antiviral activity towards CoVs, MHV-A59, bovine coronavirus strain L9 (BCoV-L9), and human enteric coronavirus 4408 (HECoV-4408) propagated in mouse astrocytoma DBT and fibroblast 17Cl-1 cells at a low micromolar concentration [EC_50_: 0.3 µg/mL] [100]. Similarly, LLC-MK2 cells, nitazoxanide and its active metabolite (tizoxanide) displayed potent antiviral activity against MERS-CoV [EC_50_: 0.92 and 0.83 µg/mL, respectively] [99]. Effective concentration of nitazoxanide against SARS CoV-2 grown in Vero E6 cells is within the range observed for other viruses [EC_50_: 2.12 μM] [45]. Current research indicates potential mechanism of antiviral activity through the induction of the interferon response via activation of protein kinase R or disruption of the unfolded protein response [101].

Clinical trials have successfully demonstrated its effectiveness in treating influenza virus, norovirus and rotavirus, hepatitis B virus, and hepatitis C virus. In a phase 2b/3 study for the outpatient management of influenza, a treatment dose of 600 mg nitazoxanide by mouth BID was associated with a 17·3hr reduction in time to alleviation of symptoms when compared with placebo [102]. In a phase 2 RCT comprising hospitalized patients with severe acute respiratory illnesses that are mainly caused by respiratory viruses, nitazoxanide failed to reduce the time to hospital discharge or the time to symptom alleviation [103]. Although nitazoxanide displayed in vitro antiviral activity towards SARS-CoV-2, prophylactic and therapeutic efficacy studies in SARS-CoV-2 animal models are necessary to determine its benefit to the clinic.

#### 5.1.11. Homoharringtonine (HHT)

Homoharringtonine is a plant alkaloid known to exhibit potent anti-viral effects against herpesviruses (Varicella Zoster Virus or VZV, HSV-1, Pseudorabies virus or PRV), coronaviruses (porcine epidemic diarrhea virus or PEDV, MHV), rhabdoviruses (vesicular stomatitis virus or VSV and rabies virus), and other viruses (hepatitis virus, Newcastle disease virus, and echovirus 1) [46]. Omacetaxine is a semi-synthetic form of homoharringtonine that is FDA-approved for treatment of chronic myeloid leukemia. It targets the phosphorylated form of eIF4E (S209), resulting in the degradation of phosphorylated eIF4E to inhibit protein translation resulting in loss of proteins (Mcl-1 and c-Myc) required for the survival of leukemia cells [46]. Interestingly, homoharringtonine has displayed antiviral effects towards SARS-CoV-2 in Vero E6 cells at a much lower effective concentration [EC_50_ 2.10 μM] than remdesivir and LPV [17]. Antiviral effect of HHT on viral infections is presumably associated with its action on phosphorylated eIF4E. Repurposing HHT for SARS-CoV-2 treatment may represent an attractive strategy and deserves further investigation.

#### 5.1.12. Emetine

Emetine is an alkaloid derived from ipecac and is FDA-approved for the treatment of ameobiasis. Emetine was demonstrated to have anti-viral effects towards a wide range of DNA and RNA viruses including bovine herpes virus-2, HSV-2, human cytomegalovirus, Buffalo Poxvirus, ZIKV, EBoV, HIV-1, Newcastle’s disease virus or NDV, pestes des petits ruminants virus, rift valley fever virus, influenza and rabies virus [46,101]. In particular, it has antiviral effects towards HCoV-OC43, HCoV-NL43, SARS-CoV-1, SARS-CoV-2, MERS-CoV, and MHV-A59 in vitro at low micromolar concentrations [46,101]. Emetine, therefore, is a broad spectrum CoV inhibitor. Emetine inhibits viral protein translation by blocking the 40S ribosomal protein S14 in host cells. Additional mechanisms of emetine include blocking HIV reverse transcriptase, inhibiting viral polymerases, trypanosomes killing through DNA intercalation and lysosomal malfunction. Emetine inhibited MERS-CoV entry into DPP4-expressing Huh7.5 cells possibly by affecting lysosomal function [101].

### 5.2. Adjunctive Therapies

#### 5.2.1. Corticosteroids

Corticosteroids are used to mitigate the host inflammatory response which contributes to acute lung injury and acute respiratory distress syndrome (ARDS) in severe pneumonia cases. However, the benefit of these agents outweighs adverse effects that include delayed viral clearance and enhanced risk of secondary infection [64]. Presently the data on corticosteroid use in SARS-CoV-1, MERS-CoV, and SARS-CoV-2 patients are inconclusive and confounding with no data from RCTs. In an uncontrolled and non-randomized study, 95/107 (88.8%) of SARS-CoV-1 patients treated with high-dose methylprednisolone recovered from progressive lung disease after the first week of illness [104]. Two studies documented evidence of possible adverse effects from corticosteroids use in SARS-CoV-1 patients [105,106]. In a randomized, double-blind, placebo-controlled trial, patients that received corticosteroids within the first week of illness documented delayed viral clearance with significantly higher viral RNA loads in plasma during second and third week of illness [105]. In another retrospective case-controlled study, patients with psychosis who also had a familial history of psychosis received higher cumulative doses of steroids than patients without psychosis [106]. While one report documented an increase in the composite endpoint of ICU admission or death [107], another report demonstrated decreased mortality in critically ill patients [108]. Among MERS-CoV ICU patients, those that received corticosteroid therapy (n = 151/309) documented delayed clearance of viral RNA with no difference in 90-day mortality after accounting for time-varying exposure [109]. Finally, a recent observational study indicated a reduced risk of mortality with the receipt of corticosteroids in SARS-CoV-2 patients with ARDS [110]. Given the adverse effects and a lack of consistent proven benefit, clinicians must carefully weigh the risks and benefits of corticosteroid use on an individual patient level [64].

#### 5.2.2. Tocilizumab and Sarilumab

Tocilizumab is an FDA-approved humanized monoclonal antibody that inhibits IL-6 receptor in membrane-bound or soluble form. It was originally approved for the treatment of rheumatoid arthritis (RA), but has been used in patient therapy of cytokine-release syndrome (CRS) following chimeric antigen receptor T cell (CAR-T) therapy [16]. IL-6 is a key driver of dysregulated inflammation that contributes to pulmonary pathology and organ damage observed in COVID-19 patients [66]. Theoretically, antibodies targeting its receptor could dampen IL-6 signal transduction and downstream inflammation, thus improving clinical outcomes. In a limited cohort of 21 COVID-19 patients, receipt of tocilizumab resulted in clinical improvement in 91% of patients as assessed by fever resolution, chest tightness relief, lung opacity resolution on CT, improved respiratory function, and no drug reactions [111]. In 56.2% of patients, lymphocyte and C-reactive protein (CRP) levels returned to normal within 5 days [111]. A recent study conducted single cell RNA-seq in patient blood cell populations isolated longitudinally at various stages beginning tocilizumab treatment (with 12 hr, 5^th^ and 7^th^ day) from two severe patients [112]. In monocytic subpopulation, a significant downregulation of processes contributing to inflammatory storm was noted upon tocilizumab treatment [112]. Like tocilizumab, sarilumab is another IL-6 receptor antagonist FDA-approved for RA treatment. Currently, RCTs of tocilizumab in combination with favipiravir (NCT04310228) and sarilumab (NCT04315298) are being tested in RCTs.

#### 5.2.3. Convalescent Plasma

The use of convalescent plasma (CP) from patients recovered from viral infections has been in practice for more than 100 years and has been used successfully for SARS-CoV-1, MERS-CoV, EboV, and H1N1 influenza [113]. Theoretically, CP obtained from recovered patients contain a large quantity of neutralizing antibodies (nAbs) capable of clearing virus and virus-infected cells [114]. In this regard, nAb titer in CP and CP therapy initiation time are critical factors that needs to be considered [114]. Firstly, in SARS- and MERS-CoV patients, the duration of nAbs is short-lived with a decline within 3-4 months after disease process. Therefore, only plasma from recently recovered patients remains to be effective for CP therapy. Secondly, in SARS-CoV patients, improved outcome was observed upon CP therapy before 14 days post onset of illness (dpoi) compared to rescue and/or salvage. In a recent pilot clinical trial comprising 10 severe COVID-19 patients given maximal supportive care and antiviral agents, one dose of CP (200 mL) transfusion (median time of transfusion: 16.5 dpoi) resulted in negative viral RNA levels, with an improvement in clinical symptoms, pulmonary function, and paraclinical criteria (increased lymphocyte counts, oxygen saturation and decreased CRP [114]. No serious adverse reactions were noted among CP treated patients [114]. These findings also point that inflammation and overreaction of the immune system that contribute to ARDS and ALI in severe patients can be alleviated by CP therapy [114].

## 6. Conclusions

In summary, SARS-CoV-2 is a highly transmissible CoV that led to the current pandemic and disruption of human activity. There is presently a limited understanding of SARS-CoV-2 pathogenesis, mostly extrapolated from SARS-CoV-1 and MERS-CoV studies. While these studies are useful, development of an animal model that faithfully recapitulates SARS-CoV-2 pathogenesis in humans will allow identification of viral and host factors critical for COVID-19. Furthermore, animal model allows researchers to carry out the immediate task of developing and testing antiviral interventions that will eventually control COVID-19 in humans. In the present summary, we detailed the current understanding of SARS-CoV-2, a result of incredible efforts of researchers worldwide.

## Figures and Tables

**Figure 1 pathogens-09-00426-f001:**
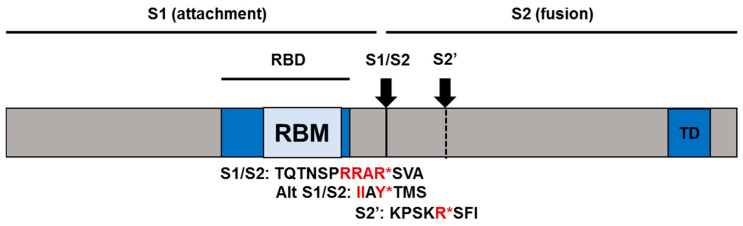
Schematic illustration of SARS-CoV-2-S; RBD: receptor binding domain; RBM: receptor binding motif; TD: transmembrane domain. Polybasic or alternative cleavage site (S1/S2), and TMPRSS2 or Cathepsin B/L (S2’) cleavage site are indicated in vertical solid and dashed lines, respectively. Amino acid sequences surrounding the protease recognition sites are indicated in red and protease cleavage site is denoted by an asterisk*.

**Figure 2 pathogens-09-00426-f002:**
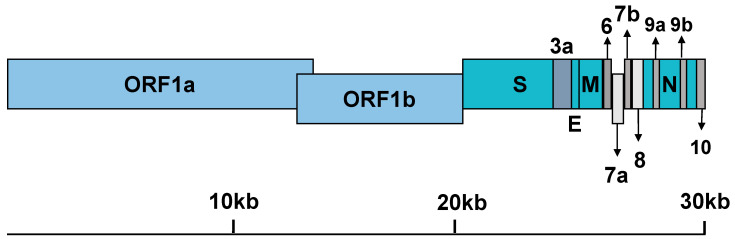
Genomic organization of SARS-CoV-2.

**Figure 3 pathogens-09-00426-f003:**
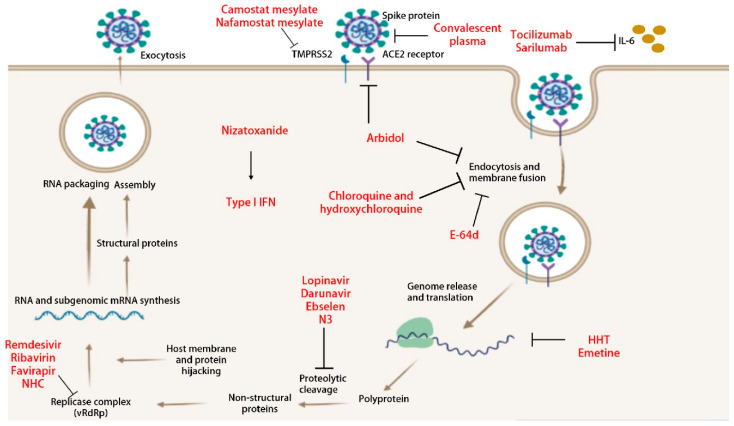
Schematic illustration of steps of virus replication and virus-induced host immune response targeted by various therapies under investigation. ACE2: angiotensin-converting enzyme 2; TMPRSS2: type 2 transmembrane serine protease; NHC: β-D-N4-hydroxycytidine; vRdRp: viral RNA dependent RNA polymerase; HHT: Homoharringtonine; and IFN: interferon.

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
