# Peer review of "A Review on SARS-CoV-2 Virology, Pathophysiology, Animal Models, and Anti-Viral Interventions"

_pathogens, 2020, doi:10.3390/pathogens9060426_

Round 1

Reviewer 1 Report

In this comprehensive review, Neerukonda et al. described essentially the different promising therapies against COVID-19. Viral entry, clinical features, and pathophysiology are discussed beforehand to provide the critical element necessary to understand how the therapies work. The reading is easy and overall, the review is very informative and complete. They listed very thoroughly the different anti-COVID-19 agent. They also provided nuances in the interpretation of the data, which is a nice touch over other reviews I have seen.

  • The stability of the virus is often overlooked in reviews, I was glad to see it there. Can the authors be more specific on the stability on a smooth surface, to stay coherent with the rest of the paragraph?
  • In section 2.4, the SARS-CoV-2-specific information is not clearly distinguished from CoV generalities. For example, the references #21 dates from 2015, well before the pandemic. Please precise and clarify in the text.
  • The section on clinical features and physiopathology was great. Some insight on the tropism of the virus tropism would be great to bridge viral entry and clinical features and pathophysiology. Also, some discussion about the actual causes of death would help put some perspective on the relevance of the different antiviral and adjunctive therapies listed after.
  • Things are moving fast: two very recent papers should now be included: Non-human primates as a potential model (Rockx et al. in Science). The paper of Yin/Xu et al. in Science can now be referenced about the mode of action of remdesivir.

Author Response

In this comprehensive review, Neerukonda et al. described essentially the different promising therapies against COVID-19. Viral entry, clinical features, and pathophysiology are discussed beforehand to provide the critical element necessary to understand how the therapies work. The reading is easy and overall, the review is very informative and complete. They listed very thoroughly the different anti-COVID-19 agent. They also provided nuances in the interpretation of the data, which is a nice touch over other reviews I have seen.

Comments

  • The stability of the virus is often overlooked in reviews, I was glad to see it there. Can the authors be more specific on the stability on a smooth surface, to stay coherent with the rest of the paragraph?

Response: Per the reviewer’s suggestion we added the stability period on smooth surfaces (2-4days; line 67 of the revised manuscript)

  • In section 2.4, the SARS-CoV-2-specific information is not clearly distinguished from CoV generalities. For example, the references #21 dates from 2015, well before the pandemic. Please precise and clarify in the text.

Response: Per the reviewer’s suggestion, we clarified it in the sub-section heading as “general features of genome replication and particle assembly; line 124 of revised manuscript)

  • The section on clinical features and physiopathology was great. Some insight on the tropism of the virus tropism would be great to bridge viral entry and clinical features and pathophysiology. Also, some discussion about the actual causes of death would help put some perspective on the relevance of the different antiviral and adjunctive therapies listed after.

Response: Per the reviewer’s suggestion, a single sentence at the beginning was added indicating tropism to the cells of respiratory and GI tracts. Specific cell types (ciliated epithelial, type II pneumocytes, intestinal epithelium etc.) were listed in animal models of COVID-19 (lines 274-276 of revised manuscript). Regarding actual causes of death, while many patients die of respiratory failure, other causes including cardiovascular failure, shock, acute kidney injury or multiple organ failure were also noted. On the other hand, COVID-19 can be just an epiphenomenon in some patients. The relative contributions of patient factors (age and comorbidities) or environmental factors (facilities, beds, personnel, or equipment etc) or genetic factors (if any) is presently not thoroughly studied or known (Vincent et al 2020). We therefore restricted our review to SARS-CoV-2 respiratory pathogenesis, viral and host factor targeting but not actual causes of death.

  • Things are moving fast: two very recent papers should now be included: Non-human primates as a potential model (Rockx et al. in Science). The paper of Yin/Xu et al. in Science can now be referenced about the mode of action of remdesivir.

Response: Per the reviewer’s suggestion, we included the non-human primate models (lines 268-285 of revised manuscript) and RDV MOA (lines 338-340 of revised manuscript) science study. Recent MA SARS-CoV-2 model is included as well (lines 216-225 of revised manuscript). Instances of small or large cats and dogs transmitted by humans were included (lines 259-267 of revised manuscript).

Reviewer 2 Report

In this report Neerukonda and Katneni review the current literature on essential aspects of SARS-CoV-2 virology. Overall, this is a wellwritten, coherent review with a lot of relevant information. Obviously, it is very difficult to write an updated review on a topic with such a rapid development. Consequently, some new information is missing. Thus, I would like to have the authors to include something on the recent development regarding vaccines, and also a few lines on the recent Remdesivir studies and their implications

Author Response

Reviewer 2

In this report Neerukonda and Katneni review the current literature on essential aspects of SARS-CoV-2 virology. Overall, this is a well written, coherent review with a lot of relevant information. Obviously, it is very difficult to write an updated review on a topic with such a rapid development. Consequently, some new information is missing. Thus, I would like to have the authors to include something on the recent development regarding vaccines, and also a few lines on the recent Remdesivir studies and their implications.

Response: We thank the reviewer for suggestions to improve the manuscript. Regarding vaccines currently in trials, we cited a review article that comprehensively reviewed the status of various vaccines (line 294 of revised manuscript). We additionally included DNA (full length S)-based immunogen that was recently tested in rhesus macaque model and a more recent VEEV-replicon based S vaccine tested in mouse-adapted SARS-CoV-2 model (lines 218-227 and 270-287 of revised manuscript). RDV MOA (lines 338-340 of revised manuscript) science study and trials leading to FDA emergency use authorization were indicated as well (lines 372-375 of revised manuscript). Some new information regarding ineffectiveness of chloroquine and hydroxychloroquine was added (lines 490-493 of revised manuscript)